



# The effects of blade structural model fidelity on wind turbine load analysis and computation time

**Ozan Gozcu and David R. Verelst**

DTU Wind Energy, Technical University of Denmark (DTU), Frederiksborgvej 399, 4000 Roskilde, Denmark

**Correspondence:** Ozan Gozcu (ozgo@dtu.dk)

**Abstract.** Aero-servo-elastic analyses are required to determine the wind turbine loading for a wide range of load cases as specified in certification standards. The floating reference frame (FRF) formulation can be used to model the structural response of long and flexible wind turbine blades. Increasing the number of bodies in the FRF formulation of the blade increases both the fidelity of the structural model and the size of the problem. However, the turbine load analysis is a coupled aero-servo-elastic analysis, and computation cost not only depends on the size of the structural model, but also depends on the aerodynamic solver and the number of iterations between the solvers. This study presents an investigation of the performance of the different fidelity levels as measured by the computational cost and the turbine response (e.g., blade loads, tip clearance, tower top accelerations CE1). The analysis is based on aeroelastic simulations for normal operation in turbulent inflow load cases as defined in a design standard. Two 10 MW reference turbines are used. The results show that the turbine response quickly approaches the results of the highest-fidelity model as the number of bodies increases. The increase in computational costs to account for more bodies can almost entirely be compensated for by changing the type of the matrix solver from dense to sparse.

## 1 Introduction

Modern wind turbine blades are large, slender and flexible composite structures with a complex pre-bent and twisted geometry. Over their operational life blades undergo large deflections and rotations due to external loads (e.g., aerodynamic, inertial and control actuator loads). An aero-servo-elastic code or framework is used to accurately calculate the complex dynamical response of wind turbines with large and flexible blades. This has led to the implementation of geometrically nonlinear structural solvers in wind-turbine-specific aero-servo-elastic codes. For example, the structural solver BeamDyn (Wang et al., 2017) was implemented in FAST (Jonkman and Buhl Jr., 2005). It uses the geometrically exact beam theory (Hodges, 1990) based on the Legendre-spectral-finite CE2 element method. Another example is a recent release of Bladed (DNV, 2016) that uses a multibody formulation (Shabana, 2013; Cardona and Géradin, 2001) to capture large structural deflections of the modeled structures. BHawC (Rubak and Petersen, 2005) is

another nonlinear aeroelastic wind turbine simulation code which uses a corotational formulation to resolve large deflections accurately.

The effect of large blade deflections on the turbine response has been studied since the early 2000s in megawatt-sized turbines. Larsen et al. (2004) performed a turbine analysis with linear and nonlinear structural solvers to investigate the effects of large blade deflections on the turbine performance. The authors concluded that the effective rotor area changes due to large blade deflections, and this alters the blade and turbine loading. In their review paper, Hansen et al. (2006) addressed the importance of nonlinear structural dynamics when large displacements occur for various wind turbines components (e.g blades, floating foundations, mooring lines). Riziotis et al. (2008) compared the blade response of first- and second-order beam models with HAWC2 (Larsen and Hansen, 2015) results. The authors concluded that the bending–torsion coupling is the main nonlinear effect for the National Renewable Energy Laboratory (NREL)

5 MW blade (Jonkman et al., 2009) and that a linear beam model underpredicts the blade torsional loads. Zierath et al. (2014) compared simulation results using different solvers with measurements of a 2.05 MW prototype wind turbine. The best agreement with measurements was obtained when a multibody dynamic solver was used, since it is able to include nonlinear effects due to large deflections. Manolas et al. (2015) investigated the nonlinear geometric effects by a comparison of different beam models of the NREL 5 MW. The authors concluded that the effects of geometric nonlinearities are still small for the NREL 5 MW turbine, but they also noted that the linear models are very close to their limit (in terms of accurately predicting the relevant deflections). Therefore, the authors recommended that future more flexible blade designs should be studied with nonlinear structural models. Beardsell et al. (2016) investigated the effects of large deflections on fatigue and extreme loads for four different wind turbines. They observed that the nonlinear effects are higher for more flexible blades and they suggested that the NREL 5 MW turbine should no longer be considered a representative of the latest generation of commercial blade designs in terms of length and flexibility. Guntur et al. (2017) compared the analysis results of various solvers with measurements of a Siemens 2.3 MW turbine. They performed the analysis using BHawC, FAST–BeamDyn and FAST–ElastoDyn. The results show that the nonlinear structural solvers (BHawC and BeamDyn), which can also model curved structures, have good agreement with measurements, while the linear solver (ElastoDyn) shows the largest discrepancy. Large blade deflections also alter the aeroelastic stability of turbines. Kallesøe (2011) showed that the coupling between the blade edgewise and torsional degree of freedom (dof) varies as a function of blade deflection shape and that the edgewise damping can decrease due to large blade deflections. Rezaei et al. (2018) showed that the blade deflections alter the damping and stiffness of the NREL 5 MW wind turbine. The authors observed that the linear models overestimate the flutter speed of the turbine.

Literature shows that large blade deflections are important to consider for a turbine response analysis, especially for long and flexible blades. The focus of the existing studies is generally limited to the blade response only, and considering a small selection of load cases. However, what is lacking is a full overview of the turbine response and a broad selection of load cases when comparing linear and nonlinear blade models without mentioning the additional computational time needed by nonlinear models. The aim of this study is to investigate the performance differences between various nonlinear blade modeling "fidelities" (in terms of number of bodies in a floating reference frame) using HAWC2. The performance of a model here is defined by its computational time and how close the loads are compared to a reference case. This is defined as the case with the highest blade model fidelity. The design load cases for power production under normal turbulence according to the International Electrotechni-

cal Commission (IEC) 61400 standard (IEC, 2005) are used for model performance comparisons. The steady turbine response of linear and nonlinear blade models is also compared in terms of power, pitch and deflection. The effect of nonlinear blade modeling on blade stability and flutter limit is investigated by considering a rotor speed runaway case. Additionally, the computational time of the two available matrix solvers (dense and sparse) in HAWC2, which uses augmented floating reference frame (FRF) formulation (Shabana, 2010), is compared.

In this study the turbine responses of DTU10MW (Bak et al., 2013) and IEA10MW (Bortolotti et al., 2019) are considered with different structural fidelity levels of the blades for 432 load cases according to design load case (DLC) CE3 1.2 (Hansen et al., 2015). Deterministic load cases (without turbulent wind) are also considered to evaluate the turbine steady-state response at various wind speeds. The loads at different points on the turbine, controller activity and turbine performance are compared. Section 2.1 introduces the solver (HAWC2) and geometrically nonlinear structure modeling in the multibody (FRF) formulation. Section 2.2 presents the reference wind turbines, load cases and their models as used for this study. Section 3 includes the calculation methods used when post-processing the results, the plots of the computation time, steady-case results, DLC 1.2 blade results, DLC 1.2 tower and performance results, stability results, and a discussion of the results. The conclusions of this study are given in Sect. 4.

## 2   Method and analysis

Evaluating the aero-servo-elastic response of large and flexible wind turbines using time domain simulations under turbulent inflow conditions requires rigorous analysis. Both the aero-servo-elastic solver and the considered model and load cases are therefore carefully outlined in the following two sections. The applied analysis method presented here is based on a numerical experiment of blades with varying structural model fidelity levels.

### 2.1   Method

The turbine analyses for the presented work were performed with HAWC2 version 12.6, which is a strongly coupled aero-servo-elastic wind turbine simulation tool. The aerodynamic solver of HAWC2 uses the blade element momentum formulation (Madsen et al., 2012; De Vries, 1979; Wilson and Lissaman, 1974) including effects of dynamic stall, dynamic inflow, wind shear on induction, tip loss, tower shadow and large blade deflections. A proportional–integral–derivative (PID) CE4 controller algorithm is used to determine the set point of the pitch bearing angle and generator torque. The servo actuators are modeled as a second-order dynamical system with an appropriate given frequency and damping. The structural dynamics of HAWC2 are based on a multi-

Wind Energ. Sci., 5, 1–14, 2020

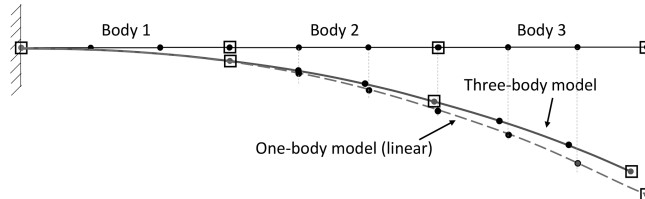

**Figure 1.** TS1 Structural modeling of a cantilever beam in floating reference system with multiple bodies, in deflected and undeflected states.

body formulation using an augmented FRF method (Shabana, 2010). Each structural element has two nodes with 6 degrees of freedom (dof) and is modeled as a linear classical isotropic or anisotropic Timoshenko beam (Kim et al., 2013). A body, defined in the FRF formulation, can be composed out of an arbitrary number of elements. Bodies are attached to each other with constraints in any of the 6 dof (three rotations and three translations). The bodies are deflected linearly, but their body reference coordinate system follows the translation and rotation from the last node of the previous body in a continuous structure model.

A general wind turbine structure can be built out of $N_e$ elements and $N_b$ bodies with constraints, but $N_b \leq N_e$. The constraints allow the user to capture the correct nonlinear geometrical response of a collection of bodies in a continuous structure as long as the deflections within one body are small (Pavese et al., 2015). In the limit case where a continuous structure model has the same number of bodies as elements ($N_b = N_e$), the solution is equivalent to the corotational approach (Krenk, 2005; Verelst et al., 2016). For example, Fig. 1 shows how the body discretization of a 2D beam structure model captures the nonlinear effect on the beam length as bending deflection occurs. The beam model has nine linear beam elements. The round markers represent the finite element nodes, and the square markers represent the body discretization of the structure. As seen in the figure, the one-body model has linear deflections with fictitious elongation due to lack of large rotations, while the three-body model shows the large rotation effects due to constraints between the bodies.

HAWC2 constructs a system of differential equations representing the equations of motion of the system with constraints (see Eq. 1) which is based on a given set of $N_e$ elements and $N_b$ bodies (Shabana, 2013) for the $i$th time step "$t_i$". TS2 $\mathbf{M} \in \mathbb{R}^{N \times N}$, $\mathbf{C} \in \mathbb{R}^{N \times N}$, and $\mathbf{K} \in \mathbb{R}^{N \times N}$ are the inertia, damping and stiffness matrices, and $N$ is the number of generalized coordinates. The generalized coordinates and their first and second time derivatives (velocities and accelerations) are shown as $\mathbf{u}$, $\dot{\mathbf{u}}$ and $\ddot{\mathbf{u}}$. Lagrange multipliers and constraint equations are represented by $\boldsymbol{\lambda} \in \mathbb{R}^{N_c}$ and $\mathbf{g} \in \mathbb{R}^{N_c}$, where $N_c$ is the number of constraints in the model. The Jacobian of constraint equations with respect to the generalized coordinates is presented by $\mathbf{G}_u \in \mathbb{R}^{N_c \times N}$. Generalized

external forces and quadratic velocity vectors, including gyroscopic and Coriolis force components, are shown as TS3 $\boldsymbol{f}$ and $\boldsymbol{f}_v$. The solver computes $\mathbf{u}$, $\dot{\mathbf{u}}$, $\ddot{\mathbf{u}}$ and $\boldsymbol{\lambda}$ at each time step for known external loads while satisfying the constraint equations. In HAWC2, the computed structural response ($\mathbf{u}$, $\dot{\mathbf{u}}$, $\ddot{\mathbf{u}}$) is sent to the aerodynamic solver. Based on these state variables, the aerodynamic solver computes the corresponding aerodynamic loads which go into the external force vector ($\boldsymbol{f}$). This load update procedure takes place at each iteration. Hence, the generalized external forces and inertia matrix are a function of time, deflections, velocities and accelerations.

$$\mathbf{M}(\mathbf{u})\ddot{\mathbf{u}}(t_i) + \mathbf{C}\dot{\mathbf{u}}(t_i) + \mathbf{K}\mathbf{u}(t_i) + \mathbf{G}_u^T(t_i)\boldsymbol{\lambda}(t_i)$$
$$= \boldsymbol{f}(\mathbf{u}, \dot{\mathbf{u}}, t_i) + \boldsymbol{f}_v(\mathbf{u}, \dot{\mathbf{u}}, t_i)\mathbf{g}(t_i) = \mathbf{0}, \mathbf{G}_u(t_i) = \frac{\partial \mathbf{g}(t_i)}{\partial \mathbf{u}(t_i)} \quad (1)$$

As the reference–rigid body ($\mathbf{u}_r$) and elastic parts ($\mathbf{u}_e$) of the generalized coordinates are separated, Eq. (1) can be written as shown in Eq. (2) for body "$j$". The stiffness and damping matrices of the body have only elastic components which are constant for linear elements. Similarly, $\mathbf{M}_{ee}$ is also constant and the constant matrices are computed once in a FRF solution process. The rest of the $\mathbf{M}$ matrix needs to be computed at each iteration together with $\mathbf{g}$, $\mathbf{G}_u$, $\boldsymbol{f}$ and $\boldsymbol{f}_v$ since they are state dependent.

$$\begin{bmatrix} \mathbf{M}_{rr}^j & \mathbf{M}_{re}^j \\ \mathbf{M}_{er}^j & \mathbf{M}_{ee}^j \end{bmatrix} \begin{bmatrix} \ddot{\mathbf{u}}_r^j \\ \ddot{\mathbf{u}}_e^j \end{bmatrix} + \begin{bmatrix} \mathbf{0} & \mathbf{0} \\ \mathbf{0} & \mathbf{C}_{ee}^j \end{bmatrix} \begin{bmatrix} \dot{\mathbf{u}}_r^j \\ \dot{\mathbf{u}}_e^j \end{bmatrix}$$
$$+ \begin{bmatrix} \mathbf{0} & \mathbf{0} \\ \mathbf{0} & \mathbf{K}_{ee}^j \end{bmatrix} \begin{bmatrix} \mathbf{u}_r^j \\ \mathbf{u}_e^j \end{bmatrix} + \begin{bmatrix} \mathbf{G}_{u_r}^{j\,T} \\ \mathbf{G}_{u_e}^{j\,T} \end{bmatrix} \begin{bmatrix} \boldsymbol{\lambda}_r^j \\ \boldsymbol{\lambda}_e^j \end{bmatrix}$$
$$= \begin{bmatrix} \boldsymbol{f}_{er}^j \\ \boldsymbol{f}_{ee}^j \end{bmatrix} + \begin{bmatrix} \boldsymbol{f}_{vr}^j \\ \boldsymbol{f}_{ve}^j \end{bmatrix} \quad (2)$$

The main driving factors in computation time of the multibody solver are the simulation time, the size of the problem (matrices) and the number of iterations. The vector $\mathbf{u}_r^j$ includes six variables to define the position and rotation of the body "$j$" reference point. The size of $\mathbf{u}_e^j$ depends on the number of elements in body "$j$". As more bodies are defined in a model, the number of generalized coordinates and state-dependent parts of the matrices increase. For example, the one-body case in Fig. 1 has 60 generalized coordinates (six reference coordinates, 54 elastic coordinates), whereas the three-body model has 72 generalized coordinates (18 reference coordinates, 54 elastic coordinates).

In HAWC2 the time integration is performed using the Newmark algorithm (Newmark, 1959) with $\beta$ and $\gamma$ constants. The updates of the current state are performed by $\triangle \mathbf{u}$ and $\triangle \boldsymbol{\lambda}$, computed according to Eq. (3). In Eq. (3) $\triangle \mathbf{r}_q$ and $\triangle \mathbf{r}_g$ are the force and constraint residuals at the current iteration step. $\mathbf{K}_{eff}$ is the effective tangent stiffness at the current state, which is shown in Eq. (4). The sparsity of the constraint Jacobian matrix ($\mathbf{G}_u$) increases with the number

**Table 1.** General properties of the reference wind turbines: DTU10MW and IEA10MW.

|  |  | DTU10MW | IEA10MW |
|---|---|---|---|
| Blade length | (m) | 86.4 | 96.2 |
| Hub radius | (m) | 2.8 | 2.8 |
| Hub height | (m) | 119 | 119 |
| Shaft tilt | (°) TS4 | 5 | 6 |
| Rotor precone | (°) | 2.5 | 4.0 |
| Blade mass | (kg) | 41 722 | 47 742 TS5 |
| Nacelle mass | (kg) | 446e5 | 446e5 |
| Prebend at the tip | (m) | 3.3 | 6.2 |
| First flapwise frequency | (Hz) | 0.61 | 0.42 |
| First edgewise frequency | (Hz) | 0.93 | 0.67 |

**Table 2.** HAWC2 turbine models' main bodies and number of elements and sub-bodies used in each main body.

| Main body name | Number of sub-bodies | Number of elements in main body |
|---|---|---|
| Tower | 1 | 10 |
| Tower top | 1 | 1 |
| Nacelle | 1 | 4 |
| Hub | 1 | 1 |
| Blade | 1–30 | 30 |

of constraints defined in the model. Different numerical approaches can be used when solving dense or sparse matrix problems. HAWC2 can optionally utilize a sparse matrix solution method in which $\triangle\boldsymbol{\lambda}$ from Eq. (3) is computed using the PARDISO sparse matrix routines (Petra et al., 2014a, b). Note that $(\mathbf{G}_\mathrm{u}\mathbf{K}_\mathrm{eff}^{-1}\mathbf{G}_\mathrm{u}^T)$ is symmetric and positive definitive for the considered HAWC2 models.

$$\triangle\boldsymbol{\lambda} = (\mathbf{G}_\mathrm{u}\mathbf{K}_\mathrm{eff}^{-1}\mathbf{G}_\mathrm{u}^T)^{-1}(\mathbf{G}_\mathrm{u}\mathbf{K}_\mathrm{eff}^{-1}\triangle\mathbf{r}_\mathrm{q} - \triangle\mathbf{r}_\mathrm{g})$$
$$\triangle\mathbf{u} = \mathbf{K}_\mathrm{eff}^{-1}(\triangle\mathbf{r}_\mathrm{q} - \mathbf{G}_\mathrm{u}^T\triangle\boldsymbol{\lambda}) \tag{3}$$
$$\mathbf{K}_\mathrm{eff} = \frac{1}{\beta h^2}\mathbf{M} + \frac{\gamma}{\beta h}\mathbf{C} + \mathbf{K} \tag{4}$$

## 2.2   Analysis

The approach in the study is based on numerical experiments of two turbines: the DTU10MW (Bak et al., 2013) and the IEA10MW (Bortolotti et al., 2019). The corresponding HAWC2 input files used for this study can be found in Gozcu and Verelst (2019). However, the versions of the DTU and IEA10MW models used here are slightly different compared to the original models: the DTU10MW has a small offset on the blade twist distribution, and the IEA10MW has a different drivetrain mass and inertia.

The properties of the blade models are shown in Table 1. It should be noted that the IEA10MW rotor has more prebend and lower blade frequencies (see Table 1), which implies a more flexible blade structure and larger geometrical couplings when compared to the DTU10MW. These differences are relevant when considering the nonlinear geometrical response of a wind turbine rotor.

It is practical to call the bodies used for a continuous structure or a component the main body, and the bodies defined in a main body are called sub-bodies. A main body can be attached to other bodies or boundaries by constraints in any direction, whereas the constraints between the sub-bodies are always in 6 dof to satisfy the continuity of the structure. In the analyses the number of sub-bodies of the blade varied from 1 (linear response) to 30 (one body for each element,

equivalent to a corotational approach). The rest of the turbine model was kept the same for a coherent comparison. The HAWC2 models of the considered turbines for this publication are composed of nine main bodies: tower, tower top, nacelle, three hubs and three blades. Table 2 shows the number of sub-bodies in the turbine models and the number of beam elements in each body. The tower, tower top, nacelle and hubs are modeled via one sub-body; in other words they are modeled as linear structures. Blades are the only parts which are modeled by multiple sub-bodies to capture large deflections. Both turbine models have 50 aerodynamic sections (or calculation points) on each blade, and the open-source Basic DTU Wind Energy controller (Hansen and Henriksen, 2013) was used. The turbulence boxes were generated by the Mann turbulence generator (Mann, 1994). A constant time step of 0.01 s was used for all considered cases. The computational time was recorded for all cases, and both the sparse and dense matrix solvers were considered.

The number of bodies in the model alters the problem size since it changes the number of generalized coordinates and constraints in the equations. The number of generalized coordinates and constraint equations can be determined by Eqs. (5)–(6). In the equations, $N_\mathrm{mb}$ is the number of main bodies, and $N_\mathrm{el}^i$ and $N_\mathrm{sb}^i$ are the number of elements and sub-bodies in the $i$th main body. The number of bodies in each blade model varies from 1 to 30. The 30-sub-body blade model (similar to the corotational model) is the most accurate with the highest $N$ and $N_\mathrm{c}$, whereas the one sub-body blade case is the linear blade case. Table 2 shows the element numbers at each main body in the turbine models. In all cases, the blades dominate the problem sizes. For example, in the one-body case the blades have 558 generalized coordinates and 18 constraint equations. For the 30-sub-body case, the three blades have 1080 generalized coordinates and 540 constraint equations. Although the problem size in the FRF formulation changes with the number of bodies defined in the model, the number of independent coordinates ($N - N_\mathrm{c}$) is always 648 for this turbine model.

$$N = \sum_{i=1}^{N_\mathrm{mb}}(N_\mathrm{el}^i + N_\mathrm{sb}^i) \times 6 \tag{5}$$

$$N_c = \sum_{i=1}^{N_{mb}} N_{sb}^i \times 6 \qquad (6)$$

The turbine analyses were carried out for steady, deterministic wind load cases, power production load cases according to DLC 1.2 and stability load cases in overspeed conditions. Steady-load cases include a full turbine model including controller under a constant wind speed with wind shear and without any yaw error. There are 23 wind speed cases starting from 4 to $26\,\mathrm{m\,s^{-1}}$ TS6 with a $1\,\mathrm{m\,s^{-1}}$ wind step size. Load cases according to DLC 1.2 include power production load cases using the normal turbulence model according to the IEC standard. In DLC 1.2 the majority of the fatigue damage of the turbine is procured over its lifetime. This can be illustrated by considering the total number of operating hours per DLC and the mean value of the respective 1 Hz equivalent loads (see Table 4). Note that the number of operating hours for DLC 1.2 is significantly larger compared to the other cases. Since the mean 1 Hz equivalent load is either similar or lower, it is safe to assume that DLC 1.2 does indeed drive the lifetime fatigue load. Consequently, only DLC 1.2 is considered for the current publication. Table 5 summarizes the simulation setup for DLC 1.2 load cases. Note that, according to the IEC standard, the use of six turbulent seeds is considered sufficient for DLC 1.2. For the analysis here 12 seeds are considered instead in order to increase the robustness of the obtained fatigue damage (Tibaldi et al., 2014) for each case with a different number of sub-bodies. In general terms further attention should be paid when comparing results from turbulent time domain simulations of nearly identical turbine models. Extreme loads can vary significantly when a large rotor is positioned slightly differently with respect to a specific temporal turbulent structure in the wind field (Natarajan and Verelst, 2012). For this analysis it can cause, potentially, large extreme load variations between the simulations of the same wind speed and seed number but a different number of sub-bodies. Such differences could be driven not by the difference in modeling (1 to 30 sub-bodies for this investigation), but by small differences in rotor azimuthal position at a specific time at which an extreme event occurs.

The stability analysis includes a turbine model which is free to speed up without generator torque and has a fixed blade pitch angle at 0° (Pirrung et al., 2014). A steady-state rotor speed under zero aerodynamic torque is found close the cut-in wind speed. From there, the wind speed is increased following a shallow linear ramp. Consequently, the rotor slowly accelerates. The instability is then determined when significant blade vibrations are observed.

## 3 Results

The simulation results of the blade models with different numbers of sub-bodies are compared to the blade with the 30-body case (highest fidelity). The loads and total number of iterations are normalized with respect to the highest-fidelity results, while the computation time is normalized with respect to the lowest-fidelity model (one sub-body, linear case) in combination with the dense matrix solver. The computation time and total iteration number are defined here as the total central processing unit (CPU) time and the sum of iterations for all load cases, respectively.

The activity of the pitch bearing is evaluated by integrating the pitch angle signal over time for all load cases; see Eq. (7). The pitch angular speed of the $j$th blade at the $i$th time step is shown by $\dot{\phi}_i^j$. There are $N_t$ number of time steps in all load cases. In addition to the total pitch angle change $\phi_{total}$, the power needed by the pitch actuator ($P_i^j$) of the $j$th blade at the $i$th time step is calculated by considering the torsion moment at the blade root ($M_i^j$) and angular speed of the pitch bearing $\dot{\phi}_i^j$; see Eq. (8). Note that the bearing friction is neglected in the equation. The max power needed by the pitch actuator might determine the size of the component (i.e., actuator, bearing).

$$\phi_{total} = \sum_{j=1}^{3} \sum_{i=1}^{N_t} \frac{\dot{\phi}_{i-1}^j + \dot{\phi}_i^j}{2} \Delta t_i \qquad (7)$$

$$P_i^j = M_i^j \times \dot{\phi}_i^j \text{ at } i\text{th time step} \qquad (8)$$

Figure 2 shows DLC 1.2 load cases' computation time and number of total iteration ratios of both turbines for dense and sparse matrix solvers. The computation time ratio is calculated with respect to the linear (one sub-body) case using the dense solver, and the ratio of the number of iterations is calculated with respect to the 30-sub-body blade case, which has the lowest number of iterations for both turbine models. The total number of iterations does not change for sparse and dense matrix solver types; therefore there is only one curve for the number of iterations. The dense matrix solver CPU time results are given only for 1-, 2-, 6-, 15- and 30-sub-body cases. The computation time is dependent on the number of iterations observed in a simulation and the number of sub-bodies of the blade. Therefore, it is possible to observe a decrease in computation time as the number of dof's and constraint equations increases. The number of iterations decreases until the 15-sub-body case, which also affects the CPU time accordingly. After the 15-sub-body case, the number of iterations remains approximately constant, and correspondingly the CPU time increases as the number of bodies increases.

The maximum dense solver computation time is observed for the 30-sub-body case. It is approximately 62 % and 70 % (see Fig. 2) slower compared to the linear case for DTU10MW and IEA10MW. Due to a sharp reduction in the number of iterations between the one- and three-sub-body cases, the computational time decreases as well, even though the complexity of the model increases. Hence, the dense solver computational cost due to the increase in model complexity increases more slowly compared to the time gained

**Table 3.** Number of generalized coordinates $N$ and constraint equations $N_c$ for the full turbine model. The number of sub-bodies refers to the sub-bodies for the different blade models; it does not refer to the total number of sub-bodies of the entire turbine.

| Blade sub-bodies | 1 | 2 | 3 | 6 | 9 | 12 | 15 | 18 | 21 | 24 | 27 | 30 |
|---|---|---|---|---|---|---|---|---|---|---|---|---|
| $N$ | 702 | 720 | 738 | 792 | 846 | 900 | 954 | 1008 | 1062 | 1116 | 1170 | 1224 |
| $N_c$ | 54 | 72 | 90 | 144 | 198 | 252 | 306 | 360 | 414 | 468 | 522 | 576 |

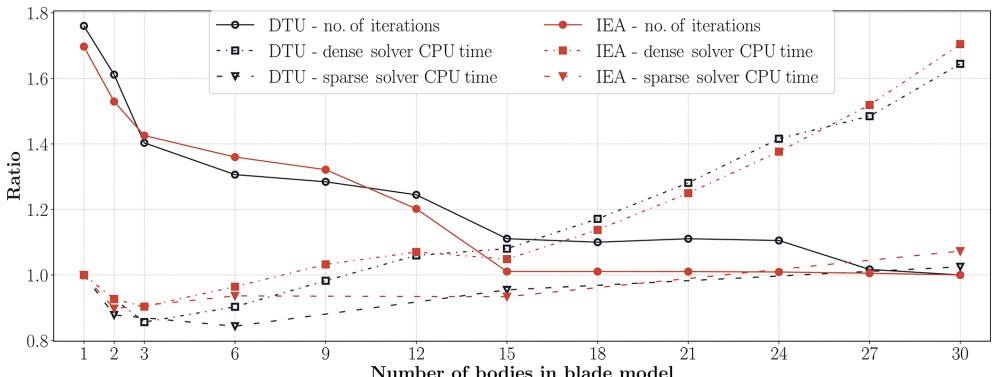

**Figure 2.** The total number of iterations is normalized by the result of the 30-sub-body case, and the total CPU time is normalized by the one sub-body dense matrix solver case for the DTU10MW and IEA10MW turbines. The CPU time ratios are given for both dense and sparse solvers.

**Table 4.** Qualitative breakdown of the fatigue load contributions of various design load cases for the IEA10MW (based on the loads reported in Bortolotti et al., 2019). Hour distribution is based on a 20-year lifetime.

| DLC | Hours | Mean blade root flapwise 1 Hz DEL ($m = 10$) |
|---|---|---|
| DLC 1.2 | 158 605 | 21 787 N·m TS7 |
| DLC 2.4 | 927 | 21 868 N·m |
| DLC 3.1 | 1528 | 20163 N·m |
| DLC 4.1 | 1528 | 17 782 N·m |
| DLC 6.4 | 4081 | 13 482 N·m |

**Table 5.** Design load cases (DLC) 1.2 power production on normal turbulence load case simulation setup.

| Simulation | Length: 600 s |
|---|---|
| Setup | Wind: 4–26 m s$^{-1}$ with steps of 2 m s$^{-1}$<br>Yaw: $-10/0/+10°$<br>Turbulence: 12 seeds per wind speed and yaw error<br>Shear: vertical and exponent of 0.2<br>Gust: none<br>Fault: none |
| TS8 Total no. of simulations | 432 |

by having fewer iterations. The number of iterations decreases only moderately between the three- and 15-sub-body cases, which is followed by a modest increase in computational time. It is only after the 15-sub-body cases, for which the total number of iterations is roughly constant, that a continuous increase in the computational time is observed as function of the number of sub-bodies. It is further interesting to note that there is no significant difference in terms of computational cost between the one- and 15-sub-body cases due to the fact that approximately 36 % and 41 % fewer iterations were observed for DTU10MW and IEA10MW, respectively.

Since the sparsity of the matrices in Eq. (3) increases with the number of bodies, the sparse matrix solver becomes computationally more efficient for models with many constraints or bodies (Dibold et al., 2007). Although not shown here, no difference was observed between the results of the dense and sparse matrix solvers. For the linear case, the CPU time is almost the same for both solver types. The sparse solver is significantly faster for the nonlinear (multibody) cases. The computational speedup for the 15-sub-body case is about 11 % and it is actually faster than the linear case with the dense matrix solver. Obtaining the sparse solution of 30-sub-body cases for the IEA turbine is about 36 % faster than using dense matrix techniques. The highest-fidelity model with a sparse matrix solver is just 9 % slower than the linear case for the IEA turbine, and this number goes down to 4 % for DTU turbine.

Wind Energ. Sci., 5, 1–14, 2020

## 3.1 Steady-wind-case results

Turbine power, blade pitch, blade effective radius change and blade tip torsion results are given for steady, deterministic wind speed conditions. Figure 3 shows the power, pitch and blade effective radius change results of linear (one body) and nonlinear (30 bodies) blade models for steady-wind-load cases. The DTU10MW results are in panel (a), while the IEA10MW results are shown in Fig. 3b. The difference between the linear and nonlinear blade models is smaller for the DTU turbine than the IEA turbine. Power differences are observed only at the below rated wind speeds CE5, whereas the pitch angles are different at the above rated wind speeds. Blade effective radius shows the blade projected to the rotation plane. Blade effective radius change differences between linear and nonlinear models are the main reasons for the pitch and power differences. The power difference reaches up to 0.3 MW at $10\,\mathrm{m\,s^{-1}}$ wind speed where the blade radius difference is around 3 m for the IEA turbine. Since the linear blade model gives longer blade effective radius, the computed turbine power is higher for the linear blade model. However, this difference does not affect the annual energy production (AEP) significantly, because the power difference is small and it occurs only at the below rated wind speeds where the power is already low. The AEP difference is less than 1 %, which is consistent with DLC 1.2 results mentioned in Sect. 3.3. The pitch angle difference between linear and nonlinear models reaches up to 0.24° at $11\,\mathrm{m\,s^{-1}}$ for the IEA turbine when the nonlinear blade model pitch is 2.79°.

Figure 4 shows blade torsion deformation results at 75 % blade span and blade tip for the linear and nonlinear blade models. Since the IEA blade is more flexible and longer than the DTU blade, the IEA torsional deflections are up to 1° larger than DTU deflections. The blade torsion deflections are large enough (up to 2.4° at IEA blade tip) to alter the turbine loads and performance. The deflections become significant in particular after the rated wind speed where the pitch activity is high. Although the torsional deformation curves of the linear and nonlinear models with respect to wind speeds look similar for the DTU10MW, the IEA10MW blade deformation curves for the linear and nonlinear models look quite different after the rated wind speed.

## 3.2 DLC 1.2 blade results

Turbine blade deflection, damage equivalent load, maximum load and cross-section load results are given for the DLC 1.2 load cases. Figure 5 shows the normalized minimum blade tip–tower clearance, maximum effective blade radius (blade tip axial position according to blade root coordinate system) and maximum edgewise deflections. The minimum tower clearance is an important design criteria, and it mostly depends on the flapwise deflection of the blades. The linear case computes lower tower clearance (larger blade deflections) than nonlinear models, and a nice approaching trend

to the highest-fidelity results is observed with an increasing number of bodies. After 15 sub-bodies the deviation from the 30-sub-body case becomes negligible. The maximum difference reaches about 5 m, which means 80 % deviation from the highest-fidelity case for the more flexible IEA turbine. There is a faster approach to the highest-fidelity results in the effective blade radius plot than the tower clearance. The IEA turbine has again a larger difference between the linear and nonlinear blade models. The diameter difference can reach up to 7 m for the IEA rotor and 1.7 m for the DTU rotor. The linear model consequently has a longer blade length than the nonlinear models due to the prebend in the blade design. The elastic part of inertia and stiffness matrices in the linear case do not change as a function of blade deflection. In other words, the linear model does not update the couplings between the various dof's as the blades deforms. The undeformed blade has a flapwise–axial displacement coupling in which the positive flapwise displacements (in the flow direction) cause an increase in blade length according to the blade root coordinate system. However, this coupling changes the sign after a certain point for the nonlinear models. The edgewise deflections computed by the linear model differ by up to 10 % compared to the nonlinear case.

Figure 6 shows the lifetime damage equivalent load CE6 (DEL) ratios between the linear (one sub-body) and nonlinear (30 sub-bodies) blade models over the normalized blade span for the DTU10MW and the IEA10MW turbines. The IEA turbine has a larger difference between linear and nonlinear cases in edgewise and flapwise DEL moments than the DTU turbine, but not so for the torsion DEL. A significant difference between the linear and nonlinear cases (30 sub-bodies) of more than 20 % can be observed for certain outboard radial stations. The flap- and edgewise DELs are consistently overestimated for the linear case, while the torsion DEL is underestimated with respect to the 30-sub-body nonlinear case.

Figure 7 shows flapwise, edgewise and torsion moment DEL ratio variations by model fidelity (number of sub-bodies in blade model) at blade stations where the maximum deviations between linear and nonlinear cases occur for each load component. The results are normalized with respect to the highest-fidelity blade model. The maximum deviation of the IEA turbine in flapwise deviation is 24 %, and it increases to 26 % for the edgewise direction. The DTU turbine has 9 % and 5 % deviations in the flapwise and edgewise directions. The results of both turbines in the flapwise and edgewise directions have a similar trend, meaning that after 15 sub-bodies the deviations become very small. The torsion DEL has the largest deviations for both turbines, and only after the nine-sub-body case can a consistent reduction in difference between the linear and nonlinear cases be observed. The deviations become quite small for cases with 27 sub-bodies or more.

Figure 8 shows the absolute maximum moment load result ratio between linear and nonlinear blade models over the

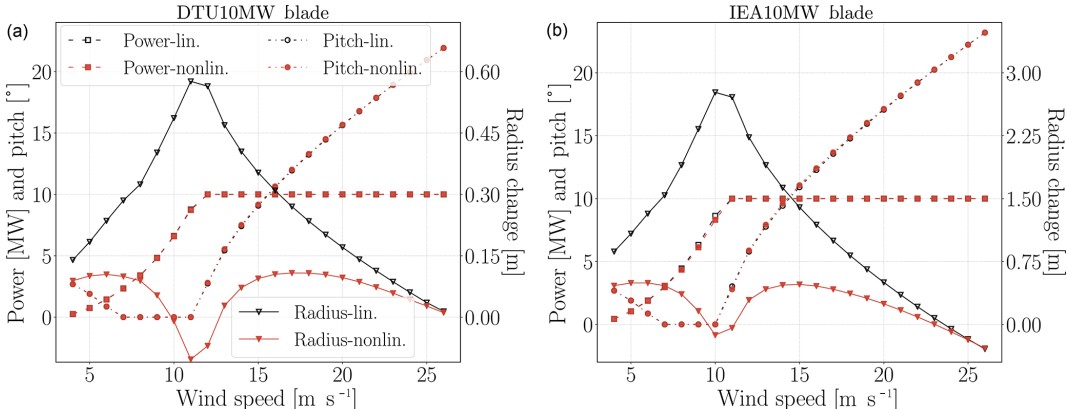

**Figure 3.** Linear and nonlinear blade model power, blade pitch (left axis in the figures) and effective blade radius change (right axis in the figures) results with respect to wind speeds for steady-wind-load cases. Panel **(a)** shows the DTU10MW results and IEA results are given in **(b)**.

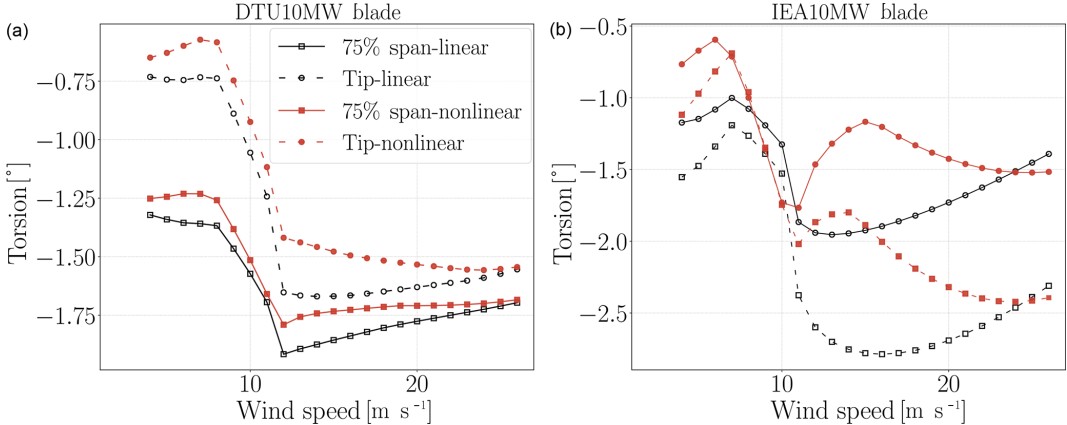

**Figure 4.** Linear and nonlinear blade model torsion deformations at 75 % blade span and blade tip with respect to wind speeds for steady-wind-load cases. Panel **(a)** shows the DTU10MW results and IEA results are given in **(b)**.

normalized blade spanwise locations. The IEA results generally have larger deviations than the DTU results. The largest difference occurs in torsion moments for both turbines. The difference in the flapwise direction reaches up to 30 % for the IEA turbine and 10 % for the DTU turbine. The edgewise deviations of both turbines reach up to 12 %. The torsion moment deviation hits 50 % in some blade regions for the IEA turbine. The torsion moments are underestimated by linear models, whereas the flapwise and edgewise moments are generally overestimated by linear models.

Alternatively, the ultimate cross-sectional loads can be visualized by considering the load envelopes. The load envelopes are the convex boundaries of the flap- and edgewise bending moment time traces considering all load cases. In doing so, the absolute magnitude and corresponding angle of the extreme loads are visualized. Figure 9 shows the cross-section flapwise and edgewise moment envelopes at blade stations where the largest deviations between linear and nonlinear cases are observed for the maximum flapwise moment load (as can be determined from Fig. 8). The largest flapwise moment deviation occurs at 43.6 and 51.1 m blade radius for the DTU and the IEA turbines. Figure 9 shows the load envelopes for 1-, 2-, 6-, 15- and 30-sub-body cases. The linear model is generally conservative with respect to the nonlinear models, and the DTU turbine has a smaller difference between linear and nonlinear blade models compared to the IEA turbine.

## 3.3 DLC 1.2 tower and performance results

Turbine tower damage equivalent load, maximum and cross-section load results are given for the DLC 1.2 load cases. The turbine performance results are also mentioned here. Figure 10 shows the normalized maximum tower top (yaw-bearing) torsion moments and maximum tower top accelerations. In the case of excessive tower top accelerations, the controller starts an emergency stop procedure. The difference in yaw-bearing torsion moment can reach up to 10 %

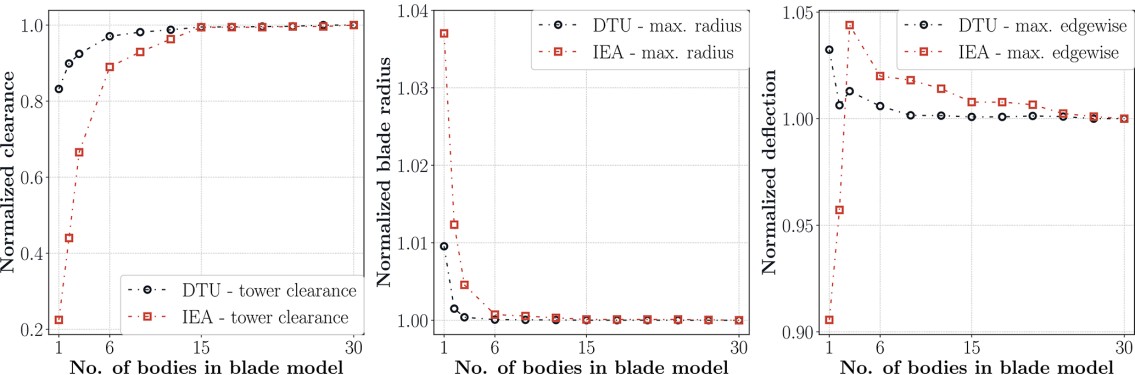

**Figure 5.** Normalized blade tip minimum tower clearance, maximum effective blade radius and blade edgewise deflection results. Values are normalized with respect to the results of the 30-sub-body case.

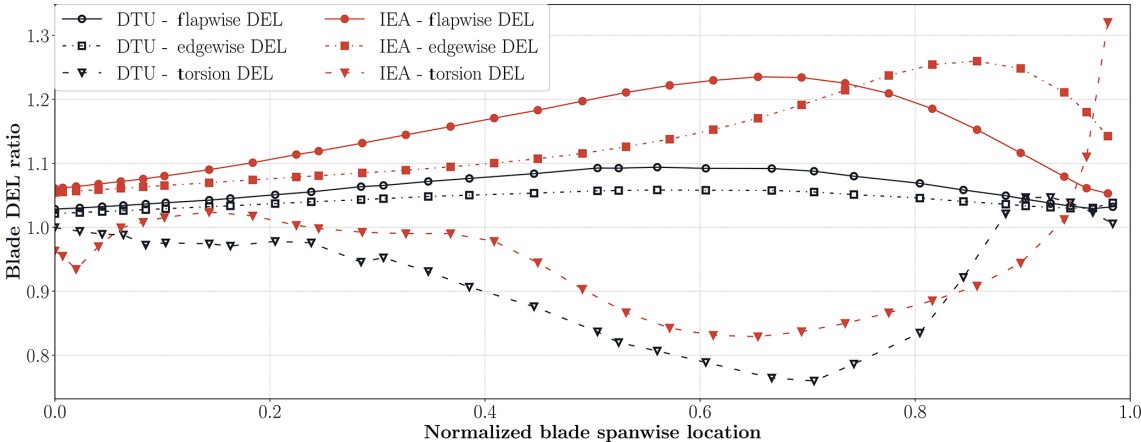

**Figure 6.** Flapwise, edgewise and torsion moment DEL ratios between linear (one sub-body) and nonlinear (30 sub-bodies) blade model over the normalized blade span for the DTU10MW and IEA10MW turbines.

for the IEA turbine. The results approach the highest-fidelity results very fast, and after the nine-sub-body case the deviations become very small compared to the 30-sub-body cases. The difference in tower top accelerations can be more than 4 % between the linear and nonlinear cases.

Figure 11 shows the DELs of the fore–aft (moment force vector perpendicular to wind direction) and side–side (moment force vector aligned with the wind direction) moments at the tower top position where the yaw actuator and bearings are located. There is a negligible deviation between the linear and nonlinear case for the side–side DEL moments for both turbine models. However, the deviations in fore–aft and torsion DELs exceed 4 % for the IEA turbine and reach 3 % for the DTU turbine. The results approach the highest fidelity model results smoothly and the deviation becomes very small after 15-sub-body cases for all channels. Figure 12 shows the tower bottom side–side and fore–aft moment load envelopes of the turbines for 1-, 2-, 6-, 15- and 30-sub-body cases. The deviations between linear and nonlinear cases are more explicit in the IEA10MW turbine than the DTU10MW turbine.

In contrast to the blade moment envelopes, the linear case is not always the more conservative approach compared to the nonlinear cases.

Figure 13 shows the normalized blade pitch actuator DEL, total pitch angle change of the turbines in all simulations computed by Eq. (7) and maximum power at pitch actuator computed via Eq. (8). The IEA turbine has a deviation of about 3 % in cumulative pitch angle results. This indicates that the controller activity is also affected by the fidelity of blade modeling. The maximum pitch actuator power depends on both blade root torsion moment and pitch angle speed. A very large deviation is observed in the pitch power results, which are 38 % and 34 % for the DTU and IEA turbines, respectively. The deviations in the IEA turbine results are generally higher than the DTU10MW turbine results; however the DTU turbine has larger deviations in terms of percentage than the IEA turbine in pitch power results. Although the highest-fidelity model causes slightly less pitch activity compared to the linear model, the actuator power increases significantly with the fidelity of the blade model. This is due

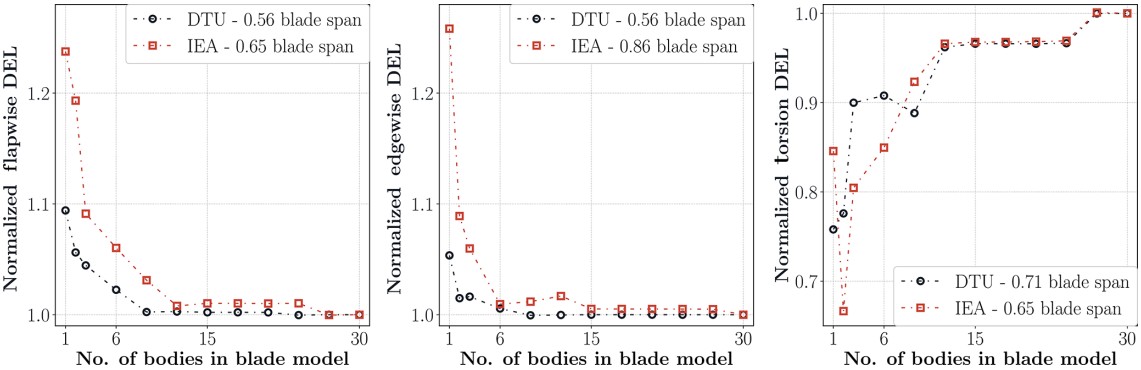

**Figure 7.** Normalized flapwise, edgewise and torsion moment DEL ratio variations with respect to the number of blade model sub-bodies at blade stations where the maximum deviations between linear and nonlinear cases occur.

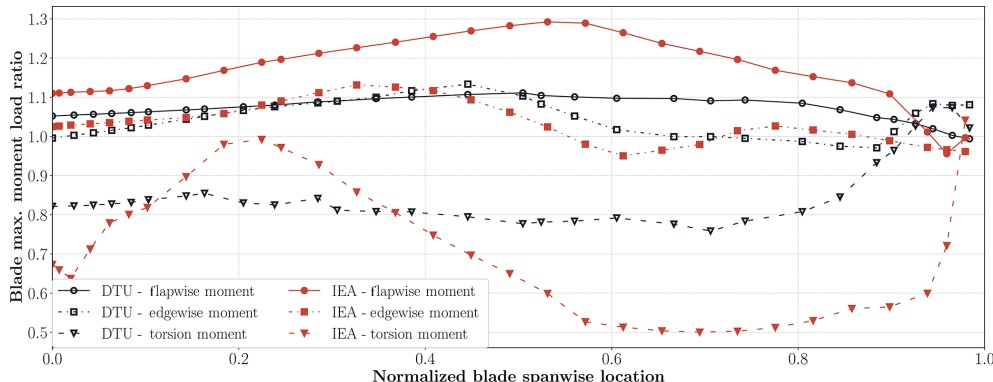

**Figure 8.** Linear and nonlinear blade absolute maximum moment load result ratio variation in DTU10MW and IEA10MW turbines with respect to blade span location.

to significantly increased blade torsional moments with increasing blade model fidelity.

The difference in annual energy production (AEP) between the different blade models is well below 1.0 %. This difference is relatively small when compared to the loads since the controller tracks the optimal operating conditions below rated wind speed and maintains the rated power above rated wind speed. Consequently, only in below rated conditions can a very small difference in power output be observed whereby the linear case results in small increase in power output compared to the nonlinear 30-sub body case.

### 3.4  Stability results

The stability of DTU and IEA turbines is evaluated by considering the linear (one body) and nonlinear (30 bodies) blade models. Blade tip torsion deformation depicts the blade vibrations and instability (flutter) clearly. Figure 14 shows the rotor speed and blade tip torsion results with respect to the wind speed. Turbines have zero aerodynamic torque at the initial wind speed, and wind speed acceleration is 0.0145 m s$^{-2}$. Results show that the DTU turbine has much higher flutter speeds than the IEA turbine for both blade mod-

els. The DTU blade linear model (blue curve) shows the flutter instability at almost the same rpm's with the nonlinear model and 1 m s$^{-1}$ higher wind speed compared to the nonlinear model (red curve). However, for the IEA linear model, flutter occurs at a wind speed which is 3.6 m s$^{-1}$ lower compared to the nonlinear model. Furthermore, the rotational speed difference between the linear and nonlinear models for the flutter instability is more than 8 rpm for the IEA blade. This shows that the linear models do not always overestimate the flutter speeds.

## 4  Discussion and conclusion

The effects of blade structural model fidelity on the turbine response, loads, stability and computation time are investigated in this study. The blades are modeled by different numbers of sub-bodies in the multibody formulation of HAWC2. The blade model geometric nonlinearity is changed from linear to the highest available fidelity level, which is equivalent to a corotational formulation. The effects of blade geometric nonlinearities are compared by exploring the results of two different blade designs with otherwise identical tower and shaft configuration. The normal power production load cases

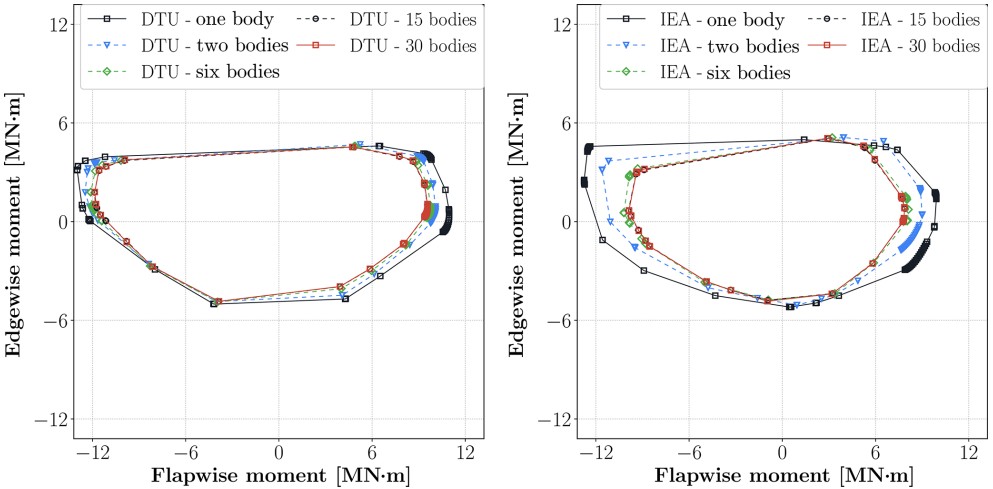

**Figure 9.** Cross-section flapwise and edgewise load envelopes at 43.6 m blade radius of the DTU turbine and 51.1 m blade radius of the IEA turbine for 1-, 2-, 6-, 15- and 30-sub-body cases.

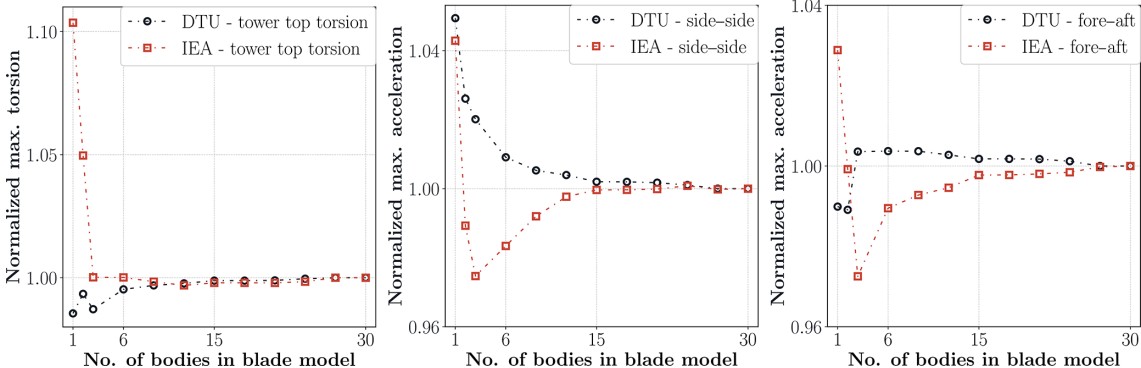

**Figure 10.** Normalized tower top maximum torsion moments, side–side and fore–aft accelerations. Values are normalized with respect to the results of the 30-sub-body case.

are selected according to the IEC 61400-1 standard (DLC1.2) but considering 12 instead of six turbulent seeds. In addition, the computational speed of the dense and sparse matrix solvers as used by HAWC2 is compared for different blade model fidelities.

CPU time can decrease by increasing the number of bodies, since the total number of aero-elastic iterations decreases as the number of bodies increases. After the total number of aero-elastic iterations becomes independent of the number of bodies, the CPU time increases by the number of bodies explicitly. The linear models have larger deflections compared to the nonlinear models and these large deflections cause larger changes in the aerodynamic forces. Consequently, the cycle between the structural response and aerodynamic forces requires more iterations for linear models. Since the sparsity of the matrices increases by the number of bodies, the sparse solver becomes more effective than the dense solver in terms of required CPU time for nonlinear problems. The geometric nonlinear effects are the most

apparent in the blade responses. The effective blade length, computed by linear and nonlinear blade models, is different by up to 3 m for steady-load cases. Hence, they have different turbine power at below rated wind speeds and different pitch at above rated wind speeds. A significant difference in blade tip–tower clearance of up to 5 m is observed, while the maximum blade tip radius can be close to 4 % higher when comparing the linear to the 30-sub-body model for DLC 1.2 load cases. The most significant differences are noted for mid- and outboard blade sections and their maximum and DEL bending moments. Depending on the blade model, the linear 1 sub-body model overestimates flap- and edgewise DELs by up to 30 %, while the torsional DEL moments are underestimated by up to 25 %. A similar trend is shown for the maximum loads: an overestimate of up to 30 % for the flap-wise extreme bending moment and an underestimated maximum torsional moment of almost 50 % when comparing the one- and 30-sub-body cases. The tower loads, however, are much less dependent on the number of blade sub-bodies.

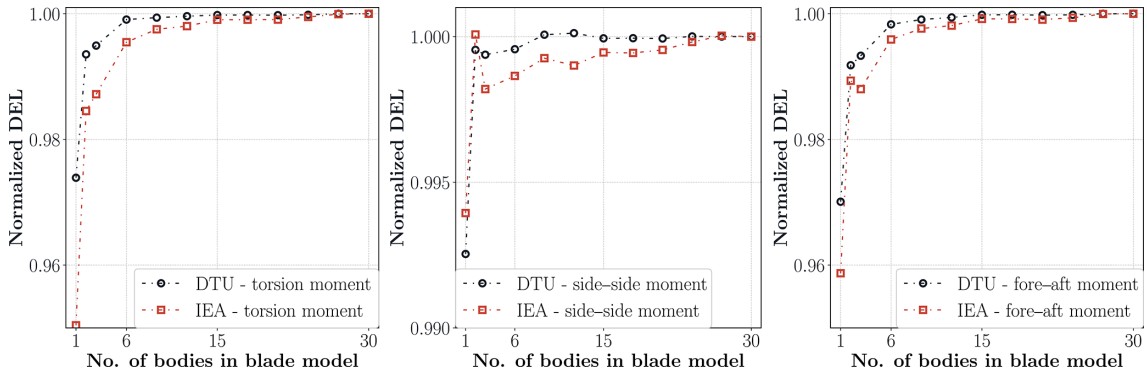

**Figure 11.** Normalized tower top torsion, side–side and fore–aft DEL moment with respect to number of sub-bodies in blade model.

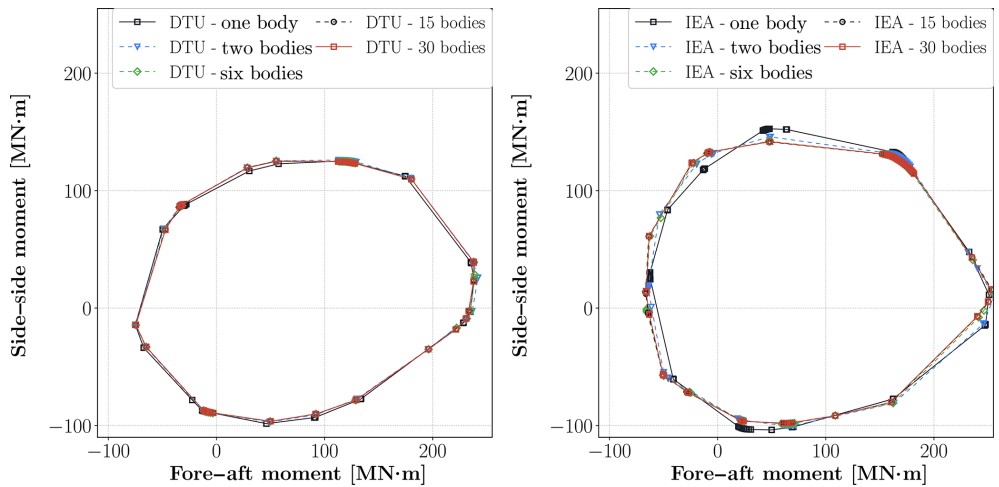

**Figure 12.** Tower bottom fore–aft and side–side load envelopes of the DTU and the IEA turbines for 1-, 2-, 6-, 15- and 30-sub-body cases.

For the tower top the largest noted differences are around 4 % for the yawing moment, but with one important distinction that fewer sub-bodies consistently underestimate rather than overestimate the loading. Further investigation is needed to understand the physical reason behind the yaw moment difference between linear and nonlinear models. The tower bottom loads are virtually unaffected as a function of blade sub-bodies. The pitch actuator maximum power is significantly underestimated by up to 30 %–40 % by the 1 sub-body blade compared to 30. The performance parameters such as power, AEP, rotational speed, thrust and shaft moment remained virtually unaffected by blade model fidelity for both steady-wind and DLC 1.2 cases. Finally, the flutter rotational speed can differ by more than 8 rpm for the linear and nonlinear blade models. Also, the linear model does not always overestimate the flutter speed.

Although there are significant differences between the linear and the nonlinear blade models (with 30 sub-bodies), the results generally approach the highest-fidelity results fast as the number of blade sub-bodies increases. In most of the studied cases the deviations in results become insignificant after 15 sub-bodies. This is also the point after which the total number of iterations does not decrease any further significantly with increasing number of sub-bodies.

The work outlined here confirms earlier studies that the nonlinear geometrical effects are significant for wind turbine blades, even more so for new turbine designs (DTU10MW vs. IEA10MW). The geometrically nonlinear effects are model dependent and are related to the size, prebend shape and flexibility of the considered blade model. The authors conclude that users are recommended to model blades with as many sub-bodies as there are structural elements, while also using a sparse matrix solver for models that have symmetric effective stiffness matrices in HAWC2. In doing so within the context of HAWC2, no increase in CPU time is noted while at the same time having the blade model with the highest structural fidelity.

**Data availability.**  TS9

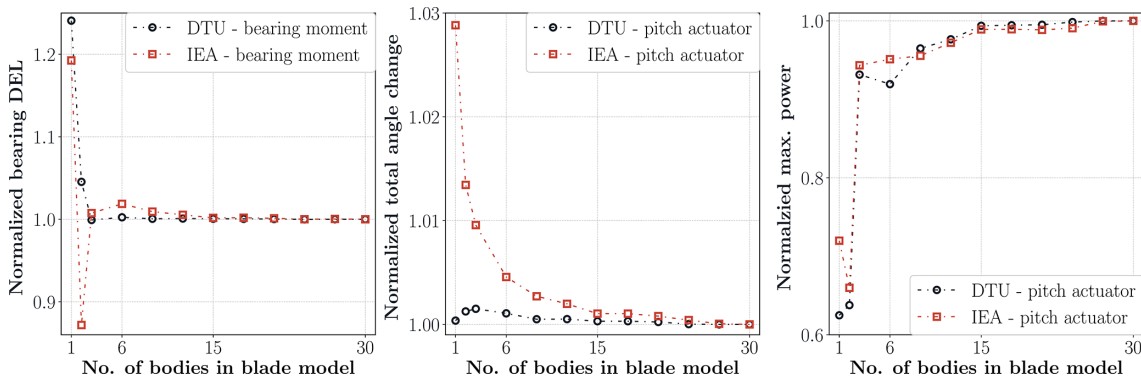

**Figure 13.** Normalized blade pitch actuator (blade root torsion moment) DEL, total pitch angle change for all load cases and maximum power at pitch actuator with respect to number of blade sub-bodies.

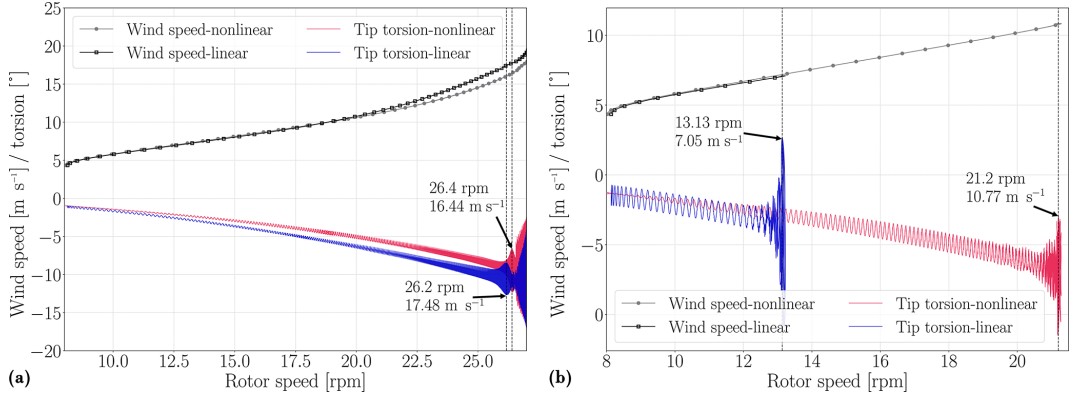

**Figure 14.** DTU10MW **(a)** and IEA **(b)** rotor speed and blade tip torsion deformation results with respect to wind speed. The flutter wind speeds are shown in the figures.

**Author contributions.** OG conducted the study as part of his PhD research. The idea was developed by OG and DRV. OG ran the analyses and performed the post-processing including results and figures with tools developed by DRV. The manuscript was written jointly by OG and DRV. CE7

**Competing interests.** DTU Wind Energy develops, supports and distributes HAWC2 on commercial terms.

**Acknowledgements.** The study was funded by Ozan Gozcu's PhD project and the HAWC2 development project in DTU Wind Energy. The authors are grateful to Sergio González Horcas and Anders Melchior Hansen for their contribution to the HAWC2 solver. Furthermore, the authors would like to acknowledge their colleagues, Sergio González Horcas, Mathias Stolpe, Suguang Dou, Riccardo Riva and Georg Pirrung for their comments on the manuscript.

**Financial support.** This research has been supported by the NAME OF FUNDER (grant no. GRANT AGREEMENT NO). TS10

**Review statement.** This paper was edited by Lars Pilgaard Mikkelsen and reviewed by two anonymous referees.

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

## Remarks from the language copy-editor

## Remarks from the typesetter

**TS9** Please provide a statement on how your underlying research data can be accessed. If the data are not publicly accessible, a detailed explanation of why this is the case is required. The best way to provide access to data is by depositing them (as well as related metadata) in reliable public data repositories, assigning digital object identifiers (DOIs), and properly citing data sets as individual contributions. Please indicate if different data sets are deposited in different repositories or if data from a third party were used. Additionally, please provide a reference list entry including creators, title, and date of last access. If no DOI is available, assets can be linked through persistent URLs to the data set itself (not to the repositories' home page). This is not seen as best practice and the persistence of the URL must be secured.