# Peer review of "The effects of blade structural model fidelity on wind turbine load analysis and computation time"

_Wind Energy Science, 2019_

## Referee Comment (RC1) · Anonymous Referee #1 · 30 Sep 2019

Dear authors, thank you for your submission. In my opinion, this is a very well written paper, with however some substantial lack of novelty. The main conclusion, modern, long, slender, flexible blades should be modeled with non-linear models that account for large deformations, is not at all a surprise and several works cited in Section 1 already go in this direction, maybe without running a comprehensive comparison as you present here, but still providing enough support to justify the claim. I have then been reflecting about possible additions to this work. One that I'd find interesting to read is an assessment of the torsional deformations of these very long blades and how this may impact the power production. You might know that blade designers fear that modern blades might suffer from substantial offsets between the prescribed pitch

angle, which is set at blade root, and the actual pitch that the sections located in the outer board see. This topic has industrial relevance and could be easily added to this work, enriching the paper. I am sure that you would see quite large differences in the torsional deformations at different number of sub-bodies... Otherwise, as said earlier, I find the paper written in a clear and concise way and I do not have any request of changes, except for the suggestion to break Section 3 - Results into at least two sub-sections 3.1, 3.2, ..., . You could for example split the analysis of rotor loads with the loads measured in the rest of the structure. I believe that this would improve readability. Best regards

---

## Referee Comment (RC2) · Anonymous Referee #2 · 20 Dec 2019

General comments: The paper deals with the influence of modelling fidelity of turbine blades on the computational time and the derived loads. This is an interesting topic and has increasingly higher attention due to the larger turbine and eventually larger deflections and thus non-linearity in the response. The paper is well written and clear. The overall purpose and approach is clearly described and easy to follow. Graphics, tables and equations are presented well and consistently. Specific comments and suggestions: The paper starts out with a good overview of previous work in the area, and among other analyses, it mentions earlier work on the influence of non-linearity on stability. This is, however, not treated further in the paper and this is an essential shortcoming. It would be interesting to use the models established in this study also

for investigating the impact on stability, e.g. blade vibrations during operation or even flutter in speed-up cases. The reference for the HAWC2 input files is not active (Page 5 line 18). In the description of the computational setup based on the IEC61400-1 standard, only the DLC1.2 is included (page 7 line 5ff). The additional effort to include all fatigue relevant cases (at least fault cases 2.4 + start-up and shut-down) would have been minor, and since these load cases often involve extreme loads and thus extreme deflections, the impact of non-linearity could be high in such cases. This is a part missing the current paper. A paragraph (page 7 line 9ff) describe how one must be careful when considering ultimate loads in cases where the statistical variation is high. The statements are correct, but what is the intension with these statements? The ultimate loads are not really covered in the analysis, apart from sparsely in Figures 8+10. Some considerations are included on the pitch activity and on the pitch power consumption. It is not mentioned or considered how the frictional moments are influencing the torsional moment and thus the power consumption of the pitch drives. This must be mentioned and if it in an easy way could be included, it will strengthen this part of the analysis. In a side remark (page 14 line 3) it is mentioned that the AEP is only influenced 1%. This does not seem consistent with the relatively large difference in effective rotor radius (page 9 line 18-19). A further detailed analysis of the power production/power curves could be included. In the analysis of the normalized maximum loads (e.g. page 10 line 6) it is unclear how the statistics are derived. Is it absolute max or average of all realizations? In the conclusion it is stated that the behavior of the yaw moment is different than other loads (fewer sub-bodies underestimate loads). What is the physics behind this? Could be correct, but it need an further analysis and an explanation should be given. It is recommended that users model blades with "as many sub-bodies as there are structural elements". How is that conclusion derived? And how many structural elements are recommended?

---

## Editor Comment (EC1) · Lars Pilgaard Mikkelsen (Editor) · 6 Jan 2020

Dear Authors, Both reviewers found your manuscript well-written and is positive regarding publication in the journal. Therefore, please go through the suggestions made by the reviewers and upload an updated manuscript for final consideration.

---

## Author Comment (AC1) · 31 Jan 2020

Dear Reviewers,

We would like to thank you for your thorough reviews and insightful comments and suggestions. We have tried to address them all, both by responding in this document, and by changing the manuscript itself. The changes in the manuscript are marked with blue.

The manuscript has also seen several minor changes to resolve grammatical issues. These changes are not marked.

Best regards,

Ozan Gözcü and David Verelst

Anonymous Referee #1

- Dear authors, thank you for your submission. In my opinion, this is a very well written paper, with however some substantial lack of novelty. The main conclusion, modern, long, slender, flexible blades should be modeled with non-linear models that account for large deformations, is not at all a surprise and several works cited in Section 1 already go in this direction, maybe without running a comprehensive comparison as you present here, but still providing enough support to justify the claim.

We agree with the reviewer that the existing studies already showed similar effects for geometrically nonlinear blade modeling. However, they don't mention the computational time effects of nonlinear blade modeling in wind turbine analysis by augmented FRF formulation and effects of sparse and dense matrix solvers on the CPU time. In addition to that, they include generally blade response rather than to turbine response for a limited number of load cases. This is mentioned in page 2 paragraph 3 (introduction section). Further, by including the IEA10MW model a much more flexible blade design is being considered for the first time in such a comparison.

- I have then been reflecting about possible additions to this work. One that I'd find interesting to read is an assessment of the torsional deformations of these very long blades and how this may impact the power production. You might know that blade designers fear that modern blades might suffer from substantial offsets between the prescribed pitch angle, which is set at blade root, and the actual pitch that the sections located in the outer board see. This topic has industrial relevance and could be easily added to this work, enriching the paper. I am sure that you would see quite large differences in the torsional deformations at different number of sub-bodies.

Thank you for the recommendation. It is actually a good idea to show torsional deformations in the manuscript. Since the DLC 1.2 results are for turbulent wind, we give statistics of the all 432 cases. However, the torsional deformations and pitch angles can be seen clearly in steady wind load case results. Hence, a figure showing torsion deformations with respect to deterministic steady wind speeds is included in section 3.1. Another figure, showing the effect of nonlinear modeling on steady power, pitch and effective blade length is also added to the manuscript.

- Otherwise, as said earlier, I find the paper written in a clear and concise way and I do not have any request of changes, except for the suggestion to break Section 3 - Results into at least two sub- sections

3.1, 3.2, ..., . You could for example split the analysis of rotor loads with the loads measured in the rest of the structure. I believe that this would improve readability.

The result section is now divided into sub-sections as suggested.

Anonymous Referee #2

General comments: The paper deals with the influence of modelling fidelity of turbine blades on the computational time and the derived loads. This is an interesting topic and has increasingly higher attention due to the larger turbine and eventually larger deflections and thus non-linearity in the response. The paper is well written and clear. The overall purpose and approach is clearly described and easy to follow. Graphics, tables and equations are presented well and consistently. Specific comments and suggestions:

- The paper starts out with a good overview of previous work in the area,and among other analyses, it mentions earlier work on the influence of non-linearity on stability. This is, however, not treated further in the paper and this is an essential shortcoming. It would be interesting to use the models established in this study also for investigating the impact on stability, e.g. blade vibrations during operation or even flutter in speed-up cases.
Thanks for the suggestion. Flutter analysis results are added to the manuscript (please see Section 3.4)

- The reference for the HAWC2 input files is not active (Page 5 line 18).
The link/DOI will be activated after the paper is published. We do this in order to avoid creating a new DOI in case small changes would have been necessary, and to have the correct DOI of the publication included in its meta-data card. Until the DOI is activated, the dataset is available via: https://figshare.com/s/b7790a9ae1efc54e8409

- In the description of the computational setup based on the IEC61400-1 standard, only the DLC1.2 is included (page 7 line 5ff). The additional effort to include all fatigue relevant cases (at least fault cases 2.4 + start-up and shut-down) would have been minor, and since these load cases often involve extreme loads and thus extreme deflections, the impact of non-linearity could be high in such cases. This is a part missing the current paper.
An explanation about the choice of the load cases is added to 2.2 Analysis section together with Table 4, which shows the importance of DLC 1.2.

- A paragraph (page 7 line 9ff) describe how one must be careful when considering ultimate loads in cases where the statistical variation is high. The statements are correct, but what is the intention with these statements? The ultimate loads are not really covered in the analysis, apart from sparsely in Figures 8+10.
 We agree that it could be left out based on the grounds you mention. However, the intention is to make the reader aware that it is always important to carefully consider that seed-to-seed variability can be significant. Since this is, in the authors opinions, not always considered carefully in general in other publications, we thought it would still be valuable to keep the statement.

- Some considerations are included on the pitch activity and on the pitch power consumption. It is not mentioned or considered how the frictional moments are influencing the torsional moment and thus the power consumption of the pitch drives. This must be mentioned and if it in an easy way could be included, it will strengthen this part of the analysis.

The pitch bearings are assumed friction-less. The manuscript was modified in away that it explicitly mention the friction-less bearing assumption in section 3 Results where the equations are given.

- In a side remark (page 14 line 3) it is mentioned that the AEP is only influenced 1%. This does not seem consistent with the relatively large difference in effective rotor radius (page 9 line 18-19). A further detailed analysis of the power production/power curves could be included.
Thank you for this review. It is actually an interesting point and now explained 3.1 Steady Wind Cases Results section. The AEP difference is about 1% for also the Steady Cases since the power difference between linear and nonlinear blade models occur only at below rated wind speeds and the difference is not very large (maximum is about 0.3 MW for IEA turbine at 10 m/s wind speed).

- In the analysis of the normalized maximum loads (e.g. page 10 line 6) it is unclear how the statistics are derived. Is it absolute max or average of all realizations?
The maximum moment forces are absolute maximum. It is mentioned in page 14 paragraph 1 (not in page 10 anymore since the structure of the results section is changed and steady state results are added).

- In the conclusion it is stated that the behavior of the yaw moment is different than other loads (fewer sub-bodies underestimate loads). What is the physics behind this? Could be correct, but it need an further analysis and an explanation should be given.
Thank you for this comment. There is no straightforward answer to this question and it needs further investigation. This is also mentioned now in the conclusion part.

- It is recommended that users model blades with "as many sub-bodies as there are structural elements". How is that conclusion derived? And how many structural elements are recommended?
The structure finite element discretization is out of the scope of the study and it deserves to be another research paper.
It is said in the conclusion section that 'The authors conclude that users are recommended to model blades with as many sub-bodies as there are structural elements, while also sing a sparse matrix solver for models that have symmetric effective stiffness matrices in HAWC2. In doing so within the context of HAWC2, no increase in CPU time is noted while at the same time having the blade model with the highest structural fidelity.' So there are two important points; first the code must use augmented FRF formulation as HAWC2 does and sparse matrix solver must be selected. Then the CPU time doesn't change significantly for the maximum body case which is the most accurate.